# Antimicrobial Stewardship during COVID-19 Outbreak: A Retrospective Analysis of Antibiotic Prescriptions in the ICU across COVID-19 Waves

**DOI:** 10.3390/antibiotics11111517

**Published:** 2022-10-30

**Authors:** Ines Lakbar, Louis Delamarre, Fanny Curtel, Gary Duclos, Karine Bezulier, Ines Gragueb-Chatti, Ignacio Martin-Loeches, Jean-Marie Forel, Marc Leone

**Affiliations:** 1Assistance Publique Hôpitaux de Marseille, Department of Anesthesiology and Intensive Care, Hôpital Nord, Aix-Marseille University, 13015 Marseille, France; 2CEReSS, Health Service Research and Quality of Life Centre, School of Medicine—La Timone Medical, Aix-Marseille University, 13006 Marseille, France; 3Assistance Publique Hôpitaux de Marseille, Department of Intensive Care Medicine, Hôpital Nord, Aix-Marseille University, 13015 Marseille, France; 4Department of Anaesthesia and Critical Care Medicine, St. James’s Hospital, D08 NHY1 Dublin, Ireland; 5Multidisciplinary Intensive Care Research Organization (MICRO), St James’s Hospital, D08 NHY1 Dublin, Ireland

**Keywords:** antimicrobial stewardship, antibiotic de-escalation, intensive care unit, Sars-CoV-2 pneumonia, bacterial superinfection

## Abstract

The demographics and outcomes of ICU patients admitted for a COVID-19 infection have been characterized in extensive reports, but little is known about antimicrobial stewardship for these patients. We designed this retrospective, observational study to investigate our hypothesis that the COVID-19 pandemic has disrupted antimicrobial stewardship practices and likely affected the rate of antibiotic de-escalation (ADE), patient outcomes, infection recurrence, and multidrug-resistant bacteria acquisition. We reviewed the prescription of antibiotics in three ICUs during the pandemic from March 2020 to December 2021. All COVID-19 patients with suspected or proven bacterial superinfections who received antibiotic treatment were included. The primary outcome was the rate of ADE, and secondary outcomes included the rate of appropriate empirical treatment, mortality rates and a comparison with a control group of infected patients before the COVID-19 pandemic. We included 170 COVID-19 patients who received antibiotic treatment for a suspected or proven superinfection, of whom 141 received an empirical treatment. For the latter, antibiotic treatment was de-escalated in 47 (33.3%) patients, escalated in 5 (3.5%) patients, and continued in 89 (63.1%) patients. The empirical antibiotic treatment was appropriate for 87.2% of cases. ICU, hospital, and day 28 and day 90 mortality rates were not associated with the antibiotic treatment strategy. The ADE rate was 52.2% in the control group and 27.6% in the COVID-19 group (*p* < 0.001). Our data suggest that empirical antibiotic treatment was appropriate in most cases. The ADE rates were lower in the COVID-19 group than in the control group, suggesting that the stress associated with COVID-19 affected our practices.

## 1. Introduction

From the onset of the coronavirus (COVID-19) pandemic, the world has faced a surge of acute respiratory infections that have often required admission to intensive care units (ICUs) and resulted in significant case fatality rates. During their ICU stay, COVID-19 patients were at high risk of hospital-acquired infections [1,2], as they were subject to invasive devices, exposed to multiple antimicrobial treatments, and potentially colonized by multidrug-resistant (MDR) bacteria [3]. In addition, the inflammatory response they experienced exposed them to a risk of relative immunosuppression [4]. Furthermore, the early stages of the pandemic were accompanied by the use of large amounts of antibiotics as prophylaxis while immunosuppressive therapy was an essential part of the therapeutic armamentarium in COVID-19 patients, thus potentially leading to an increase in bacterial infections [2,5].

MDR bacteria acquisition, triggered by excessive use of antibiotics, is a growing threat to public health [6]. The acquisition of MDR bacteria is associated with an increased risk of poor clinical outcomes and death [5,6,7]. In the fight against antimicrobial resistance, the concept of antimicrobial stewardship is well-established. One of the most concise definitions [8] might be “a coherent set of actions which promote using antimicrobials responsibly” based on the premise that antimicrobial treatments are a limited resource and that inappropriate use can have serious adverse effects on the patient, such as increased morbidity and mortality. Antimicrobial stewardship programs have a central role in optimizing antibiotic prescription both initially (i.e., empirical and de-escalated) and secondary adapted [8,9,10].

While the demographics, clinical features, and outcomes of ICU patients admitted for a COVID-19 infection have already been characterized in extensive reports from several parts of the world [11,12,13], little is known about antimicrobial stewardship for those patients. The COVID-19 outbreak has caused a major disruption to healthcare systems and practices and during the height of the pandemic, many antimicrobial stewardship resources were reallocated to meet the needs of the healthcare workforce, limiting the time they could devote to antimicrobial stewardship initiatives and recommendations for the rational use of antibiotics in COVID-19 patients [14]. This reprioritization of human resources toward clinical needs and the scarcity of clinical resources during the pandemic created the potential for inappropriate antimicrobial prescribing [5,15,16]. We designed this study to investigate our hypothesis that the COVID-19 pandemic disrupted our routine practices of antimicrobial stewardship and therefore has likely affected the rate of antibiotic de-escalation (ADE), patient outcomes, infection recurrence, and MDR bacteria acquisition.

## 2. Results

### 2.1. Patient Characteristics

During the study period, 593 patients were admitted to the ICUs for acute respiratory failure due to COVID-19. Among them, 170 received antibiotic treatment for a suspected or proven superinfection, of which 141 treatments were empirical (Figure 1). The age was 63.3 (±12.6) years in the overall cohort, which was composed mainly of males (72.9%). The Charlson score was 4.59 (±2.28), and the SAPSII and SOFA scores were 41.1 (±15.7) and 5.1 (±2.9), respectively. One hundred thirty-eight (81.2%) patients received dexamethasone as part of their COVID-19 bundle of treatment (Table 1). Only 21 (12.4%) patients did not undergo mechanical ventilation during their ICU stay. Seventy-one (46.5%) patients required vasopressors (Table 1).

Bacteria responsible for infection were identified in 55 (39%) of the 141 patients. The most common site of infection was the lung (88%), and the most frequently identified bacteria was *Staphylococcus aureus*. Empirical antibiotic treatment consisted of a beta-lactam in most of the cases (98.5%, piperacillin/tazobactam 67.4%) and was a combination of antibiotics in 35.4% of the cases. The most frequent combination was beta-lactam plus aminoglycosides. Details regarding the site of infection, the causative bacteria, and the type of empirical antibiotic treatment are provided in Appendix A.

### 2.2. Primary Outcome

Among the patients receiving an empirical antibiotic treatment, the treatment was de-escalated in 47 (33.3%) patients, escalated in 5 (3.5%) patients, and continued in 89 (63.1%) patients (Figure 1). The ADE consisted of the interruption of one or more components of the empirical antibiotic treatment in 18 (38.2%) patients or the replacement of beta-lactam with a narrower-spectrum antibiotic in 29 (61.7%) patients (Table 2). We found that ADE was not performed in 8 (8.9%) out of 89 patients in the continuation group where data suggested that ADE was feasible. There were no significant differences between ADE, escalation, and continuation groups regarding their baseline characteristics (comorbidities, severity scores, administered treatment during ICU stay) except for the severity score at ICU admission and the need for vasopressors (Table 1). ADE rates did not significantly differ between the COVID-19 waves (*p* = 0.11).

### 2.3. Secondary Outcomes

#### 2.3.1. Empirical Antibiotic Treatment Appropriateness in the COVID-19 Group

Among the 55 patients with a documented infection, the empirical antibiotic treatment was appropriate, i.e., covering the identified bacteria, in 50 (90.9%) patients. The five patients who received inappropriate antibiotic treatment were in the escalation group. The rate of appropriate empirical antibiotic treatment did not differ across COVID-19 waves (*p* = 0.68).

#### 2.3.2. Antimicrobial Resistance in the Overall COVID-19 Group

In all, nine (0.05%) out of 170 patients acquired antimicrobial resistance during their ICU stay, as attested by the results of rectal swabs or other samples (Table 1). Patients acquired antimicrobial resistance more frequently during the first wave than during the second and third waves (*p* < 0.01). There was no association between antibiotic prophylaxis rate (*n* = 68, *p* = 0.15), empirical antibiotic treatment rate (*n* = 141, *p* = 0.36), antibiotic strategy (ADE, escalation, or continuation) (*p* = 0.11), or hospital length of stay before ICU admission (*p* = 0.24) and overall antimicrobial resistance acquisition.

#### 2.3.3. Clinical Outcomes in the COVID-19 Group

Antibiotic treatment duration was 10.2 (±10.6) days in the ADE group, 12 (±5.3) days in the escalation group, and 4.8 (±3.0) days in the continuation group (*p* < 0.001) (Figure 2). Excluding patients who discontinued all antibiotics, the mean duration of antibiotic treatment in the continuation group increased to 7.3 (±2.8) days (*n* = 41). The rates of recurrence were 27.7%, 60.0%, and 9.0% in the ADE group, escalation group, and continuation group, respectively (*p* = 0.001). Similarly, the rates of relapse were higher in the escalation group (*p* = 0.009) (Table 1).

There was no difference between groups regarding the number of days of mechanical ventilation (*p* = 0.054), ICU length of stay (*p* = 0.18), and hospital length of stay (*p* = 0.056) (Figure 2). The ICU, hospital, and day 28 and day 90 mortality rates were not associated with the antibiotic treatment strategy (ADE, escalation, or continuation) (Table 1, Figure 2).

### 2.4. Control Group

Sixty-seven patients were included in the control group (Appendix A; Appendix A). The main differences between the two groups were the body mass index (BMI) (30 ± 5.8 in COVID-19 vs 24.5 ± 4.9 in control group, *p* = 0.001), previous hospitalization (11.2% in COVID-19 vs 49.3% in control group, *p* = 0.01) and previous antibiotic treatment in the 90 days (27.1% in COVID-19 vs 46.3% in control group, *p* = 0.001) (Appendix A). The control group had higher SAPSII than the COVID-19 group (49 ± 16 vs. 41 ± 16, *p* < 0.001) (Appendix A). Among those receiving an empirical antibiotic treatment (*n* = 58), empirical antibiotic treatment was de-escalated, escalated, and continued in 34 (58.6%), 3 (5%), and 21 (36.2%) patients, respectively (Appendix A).

The ADE rate was 52.2% in the control group and 27.6% in the COVID-19 group (*p* < 0.001). The rates of acquired resistance were higher in the control group than in the COVID-19 group (16.4% vs. 5.3%, *p* = 0.009). No differences were reported for the ICU, day 28 and day 90 mortality rates between the two groups (Appendix A).

## 3. Discussion

To our knowledge, this is the first study on ADE in ICU COVID-19 patients, which was conducted over an extensive period including four waves of COVID-19 patients. We found an ADE rate of 33.3% that did not significantly differ across the successive waves of patients. Antibiotic treatment duration in the ADE and escalation groups was prolonged compared with that of the continuation group, but this finding did not translate into significant changes in mortality rate or duration of ICU or hospital stay. Interestingly, only 8.8% of the patients in the continuation group were eligible for ADE, while 87% of empirical antibiotic treatments targeted the bacteria responsible for infection. Finally, we found that antibiotics were less de-escalated in the COVID-19 patients than in the control group.

Regarding ADE rate, our results are consistent with previous studies involving non-COVID-19 patients. In a systematic review of 14 studies on ADE in ICU patients, Tabah et al. found rates between 32% and 81% [18]. De Bus et al. conducted the largest prospective study by including 1495 ICU patients worldwide and found a 16% de-escalation rate [19]. However, the ADE rate was higher in the control group than in the COVID-19 group. Within the limitations due to baseline features between the two groups, this finding may suggest that the workload due to COVID-19 and the disruption related to the pandemic affected our practices in terms of antimicrobial stewardship. Previous studies have reported higher rates of ADE, but this can be also due to variations in the definition of ADE or the inclusion of less critically ill patients [18].

De-escalation, escalation, or continuation of empirical antibiotic treatment did not affect mortality rates. This finding is in line with previous studies [20,21]. In contrast, the duration of antibiotic treatment differed according to the occurrence of ADE, escalation, or continuation. A prolonged treatment duration was associated with ADE in a randomized controlled trial that compared ADE and continuation of “pivotal” empirical antibiotics [21], as well as in patients with neutropenia [22]. The increased rate of recurrence in the ADE and escalation groups also confirmed previous findings [21]. Decreased intensivist attention to patients receiving narrow-spectrum antibiotics, because of reduced costs or supposed ecological effects, may be an explanation.

The effect of ADE on the emergence of MDR bacteria is a matter of debate. In our cohort, only a few patients developed MDR bacterial infections, which limits the impact of this finding. In an observational study, de Bus et al. did not find associations between ADE and the emergence of MDR pathogens [19]. Interestingly, it has been suggested that a single day of antibiotic treatment can alter the microbiome [23]. Hence, ADE occurring around day 3 probably has only a weak effect on resistance levels. The development of rapid diagnostic tests, such as multiplex PCR, may affect the occurrence of resistance to a greater extent than ADE by reducing the global use of antibiotics [24]. In addition, the relationship between the spectrum of antibiotics and their potential regarding the emergence of resistance remains unclear and may depend on several variables, e.g., the route of elimination of each drug [25]. Finally, other factors may be involved in the emergence of MDR bacteria, notably immunosuppression. Our data showed a higher ADE rate and unexpectedly a higher rate of acquired resistance in the control group than in the COVID-19 group. This could be explained by differences in baseline features between the two groups, particularly in terms of MDR risk factors (previous hospitalization, previous antibiotic treatment) and severity scores upon ICU admission, reflecting higher rates of frailty and exposure to MDR in the control group. Hence, it is urgent to implement antimicrobial stewardship programs in all hospital training programs and during medical education [26].

There are limitations to our study. First, although we included patients from three ICUs, the study was conducted at a single institution and thus requires extrapolation to other hospitals to be validated. Second, our study has limitations inherent to its retrospective design, although data were collected for a short period. Of note, we did not use a contemporary control group, but our local practices have been assessed elsewhere [27]. Indeed, during the pandemic, the number of non-COVID-19 patients was not sufficient to provide a statistically relevant comparison. However, we assessed the rate of ADE in a historical control group consisting of patients admitted to the three ICUs the year before the pandemic. Third, the patients whose antibiotics were stopped at the time of reporting microbiological results were classified in the continuation group, as the guidelines stated that early discontinuation of all antibiotics cannot be considered ADE [28]. This could have led to bias in our analysis. Fourth, other antimicrobial agents, such as antifungals and antivirals were not included in our analysis. Finally, we did not assess the doses of antibiotics, although our local protocols are in line with guidelines, including high doses, continuous infusion, and therapeutic dose monitoring, if feasible [29].

## 4. Materials and Methods

### 4.1. Study Design and Population

We conducted a retrospective, observational study designed to review the prescription of antibiotics in three ICUs during the COVID-19 pandemic. All adult patients with a SARS-CoV-2 infection confirmed by real-time reverse transcriptase polymerase chain reaction (RT-PCR) of nasopharyngeal samples upon ICU admission to one of the three ICUs located on the site of the North Hospital of Marseille were screened. Among these, only patients with at least one episode of clinically suspected or proven bacterial superinfection were included. Exclusion criteria included an ICU stay shorter than two days, the need for extracorporeal membrane oxygenation (ECMO), and a history of lung transplant.

Our study covered the consecutive waves of COVID-19 patients requiring ICU admission, extending from March 2020 to December 2021. We identified four waves of admissions to the ICU during the study period: (1) from 1 March to 31 May 2020, (2) from 1 June to 30 December 2020, (3) from 31 December 2020 to 30 June 2021, and (4) from 1 July to 31 December 2021.

The three ICUs initially contained 39 beds total, but the number of beds increased to 62 during the different waves. In the ICUs, antibiotic management was discussed for each admission to the ICU and during each daily round. Only senior intensivists were allowed to prescribe antibiotic treatment. A weekly meeting with microbiologists was set up to discuss the anti-infective treatment of patients, including during COVID-19 waves.

### 4.2. Extracted Data

For each included patient, demographic, clinical, and outcome data were extracted from electronic medical records in the three participating ICUs. Confidentiality was ensured by anonymously capturing patient data and storing it in a password-protected file.

The extracted data were as follows: demographical data at admission, including age, sex, Charlson comorbidity index score, and other comorbidities (obesity, arterial hypertension, diabetes, smoking, lung or heart or cerebrovascular or renal or liver or connective tissue diseases, cancer, and dementia), emergency room management and antibiotic prescription, and severity scores upon ICU admission (Sepsis-related Organ Failure Assessment (SOFA) and Simplified Acute Physiology Score II (SAPSII)).

Further data were collected after the onset of suspected or proven superinfection, such as SOFA and SAPSII, clinical signs and symptoms, the need for ventilatory support, radiology and laboratory findings, antibiotic administration, results of blood cultures, and exposure to an immunosuppressive treatment during the ICU stay before the superinfection episode. At discharge, the number of days of mechanical ventilation, length of ICU and in-hospital stays (in days), and date of death if death occurred either in the ICU or during the hospital stay or after discharge (data reported from the French national database of deaths) were collected. Immunosuppressive treatments at ICU admission were defined as the administration of either corticosteroids or any immunosuppressive treatment in the context of solid organ transplantation before the index COVID hospitalization. Immunosuppressive treatment administration during hospitalization was defined as follows: administration of anti-interleukin (IL)-1 (anakinra) or anti-IL6 (tocilizumab) or anti-JAK2 (ruxolitinib) or/and high-dose corticosteroids (as described elsewhere [17]). Dexamethasone administration was not considered an immunosuppressive treatment. In the case of multiple episodes of superinfections, only the first one was considered in our analysis.

### 4.3. Definitions

Superinfection was defined as any suspected or proven bacterial infection requiring the use of an antibiotic treatment. Proven bacterial infections were then classified according to international guidelines [30] (Appendix A). In case of multiple bacterial infections during the ICU stay, only the first one was considered.

Antibiotic prophylaxis consisted of the administration of prophylactic antibiotics at ICU admission before the suspicion of bacterial infection (this prophylaxis mainly combined a third-generation cephalosporin with azithromycin and was mostly prescribed during the first wave). It was intended to prevent potential bacterial superinfection during SARS-CoV-2 infection and was systematically administered on admission to ICU to all COVID-19 patients in the first wave. This practice was gradually discontinued over the waves. Empirical antibiotic treatment was considered the use of broad-spectrum antibiotics while awaiting results for biological samples from patients suspected of bacterial superinfection according to international guidelines [31]. Targeted antibiotic treatment was defined as the use of an appropriate antibiotic treatment according to the antibiotic susceptibility test of the causative bacteria. Patients who received targeted antibiotic treatment as first-line therapy were classified in the “no empirical treatment” group. Patients who received both empirical and secondarily targeted antibiotic treatment were classified in the “empirical treatment”.

ADE was defined in accordance with international guidelines as the replacement of broad-spectrum antibiotics with a narrower-spectrum or a lower ecological impact antibiotic, or the discontinuation of components of an antibiotic combination [28]. Antibiotic escalation was defined as the broadening of the antibiotic spectrum after the identification of the causative bacteria and its antibiotic susceptibility test. The third strategy was defined as “continuation” when empirical antibiotic treatment was neither escalated nor de-escalated, as described elsewhere [20]. Antibiotic treatments were classified according to the first change in treatment (or lack thereof) between day 0 and the day the microbiological results were returned.

Finally, relapse of infection was defined as the occurrence of the same organ dysfunction due to the same bacteria responsible for the previous episode, and recurrence was defined as the occurrence of the same organ dysfunction but due to different bacteria than reported in the previous episode.

Bacteria were defined as MDR following the method described elsewhere [32]. MDR bacteria acquisition was searched for in all microbiological samples, including rectal swabs taken on a weekly basis as part of routine screening (these tests were repeated regardless of whether the first or the following rectal swabs were positive). Overall acquired resistance was defined as the appearance of an MDR bacteria either in rectal swabs or other samples. Patients who were already positive for MDR on admission were not considered to have acquired antimicrobial resistance unless a new type of resistance or new MDR bacteria emerged during their stay in the ICU.

### 4.4. Outcomes

Our primary outcome was the rate of ADE in patients receiving antibiotics for a suspected or proven superinfection. Secondary outcomes included variables describing the use of antibiotic treatment, such as the rate of empirical antibiotic treatment, whether it was effective against the causative bacteria, and the rate of acquired antimicrobial resistance during the ICU stay. We also included variables comparing ADE and non-ADE patients during their ICU stay: antibiotic treatment duration, infection relapse and recurrence rates, rates of mechanical ventilation, hospital and ICU length of stay, in-hospital mortality rates, and mortality rates at 28 and 90 days. Finally, further analyses were performed to search for differences between waves for all the pre-defined outcomes.

### 4.5. Control Group

To compare antibiotic stewardship during pandemics with antibiotic stewardship prior to pandemics, we decided to gather data regarding the rates of ADE, acquired antimicrobial resistance and mortality rates over a three-month period from December 1st, 2018 to February 28th, 2019 in ICU patients treated for any suspected or proven infection. All adult patients admitted to one of the three ICUs during the aforementioned period were screened for inclusion criteria. Among these, only patients with at least one episode of clinically suspected or proven bacterial infection were included. Exclusion criteria included an ICU stay shorter than two days, the need for extracorporeal membrane oxygenation (ECMO), and a history of lung transplant.

### 4.6. Ethical Considerations

The study was approved by the Committee for Research Ethics of French Society of Anesthesia & Intensive Care Medicine (IRB 00010254–2021–207). Each patient included in this study was contacted with a written letter. Patient consent for the use of their anonymized data was considered granted when a positive written response was received in return or if no response was given, according to French law [33].

### 4.7. Statistical Analysis

No sample size was calculated a priori. Numerical variables are expressed as mean (SD) or median (Q1-Q3), while categorical variables are expressed as numbers (%). Missing values were displayed for each variable, when present, in the table describing the patients’ characteristics. No imputation of missing data was planned or performed.

Comparisons of numerical variables between groups were performed via a Mann–Whitney nonparametric test in cases of two groups, or Kruskal–Wallis followed by Dunn’s test (with Bonferroni correction) in cases of more than two groups. Categorical variables were compared through chi-squared or Fisher exact test. To explore the role of ADE in 28-day mortality, we performed bivariate comparisons of several potential risk factors for 28-day mortality prior to including factors significantly associated with this outcome in multivariate logistic regression (i.e., *p*-value < 0.05). A *p*-value of <0.05 was required for statistical significance, and tests were two-tailed.

The statistical analysis was performed using R software 4.0.4 (R Core Team, R Foundation for Statistical Computing, Vienna, Austria, 2020). The STROBE checklist was used to report the results of the present study [34].

## 5. Conclusions

In conclusion, the results of our retrospective study suggest that empirical antibiotic treatment was appropriate in approximately 87% of patients with COVID-19. The rates of antibiotic de-escalation, escalation, and continuation were 33.3%, 3.5%, and 63.1%, respectively. The ADE rates were lower in the COVID-19 group than in the control group, suggesting that the stress associated with COVID-19 affected our practices. The ADE and escalation groups were associated with increased durations of antibiotic treatment and enhanced rates of recurrence, but these did not translate into prolonged ICU stays or increased mortality rates.

## Figures and Tables

**Figure 1 antibiotics-11-01517-f001:**
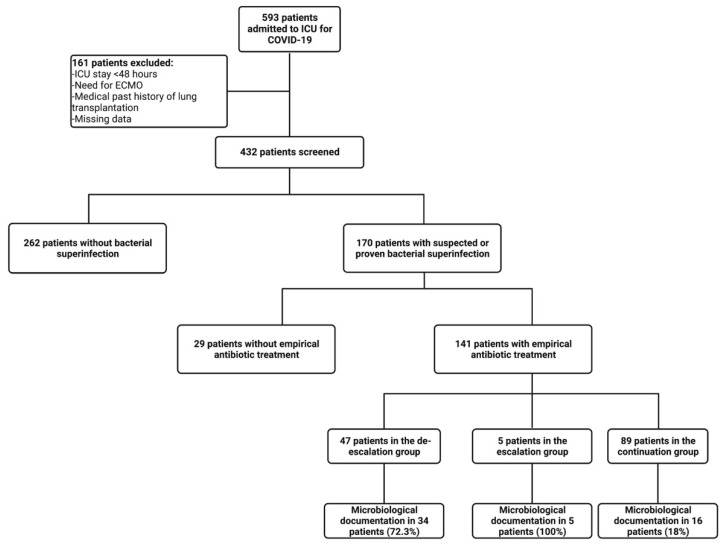
Flow chart of the study.

**Figure 2 antibiotics-11-01517-f002:**
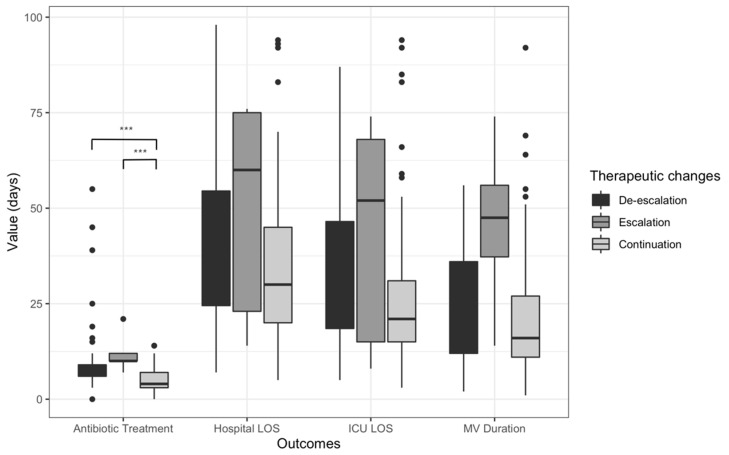
Graphic representation of the duration of antibiotic treatment, ICU and hospital length of stay, and mechanical ventilation according to antibiotic treatment strategy in the COVID-19 group. ***: *p* = 0.001.

**Table 1 antibiotics-11-01517-t001:** Patients characteristics.

	Overall (Superinfected Patients) (*n*= 170)	No Empirical Antibiotic Treatment (*n* = 29)	Empirical Antibiotic Treatment (*n* = 141)			
			De-escalation (*n* = 47)	Escalation (*n* = 5)	Continuation (*n* = 89)	*p*-Value
**Age**						
Mean (SD)	63.3 (12.6)	63.2 (14.6)	63.5 (9.51)	62.6 (7.47)	63.3 (13.7)	0.548
Median [IQR]	66.0 [57.0–72.0]	68.0 [58.0–74.0]	63.0 [56.5–69.5]	65.0 [58.0–68.0]	67.0 [57.0–74.0]	
**Gender**						
F	46 (27.1%)	14 (48.3%)	8 (17.0%)	1 (20.0%)	23 (25.8%)	0.5
H	124 (72.9%)	15 (51.7%)	39 (83.0%)	4 (80.0%)	66 (74.2%)	
**BMI**						
Mean (SD)	30.0 (5.84)	29.9 (6.67)	30.4 (6.24)	27.9 (4.76)	30.0 (5.46)	0.53
Median [IQR]	29.0 [26.0–33.4]	29.0 [26.4–31.5]	28.9 [26.4–35.0]	26.8 [26.0–27.0]	29.3 [26.0–33.1]	
Missing	11 (6.5%)	2 (6.9%)	5 (10.6%)	0 (0%)	4 (4.5%)	
**Obesity**						
0	91 (53.5%)	17 (58.6%)	24 (51.1%)	4 (80.0%)	46 (51.7%)	0.473
1	71 (41.8%)	11 (37.9%)	18 (38.3%)	1 (20.0%)	41 (46.1%)	
Missing	8 (4.7%)	1 (3.4%)	5 (10.6%)	0 (0%)	2 (2.2%)	
**Immunosuppression ***						
0	149 (87.6%)	26 (89.7%)	40 (85.1%)	5 (100%)	78 (87.6%)	0.626
1	21 (12.4%)	3 (10.3%)	7 (14.9%)	0 (0%)	11 (12.4%)	
**Invasive procedures as outpatients**						
0	169 (99.4%)	29 (100%)	47 (100%)	5 (100%)	88 (98.9%)	0.745
1	1 (0.6%)	0 (0%)	0 (0%)	0 (0%)	1 (1.1%)	
**Previous hospitalization < 90 days**						
0	151 (88.8%)	26 (89.7%)	42 (89.4%)	5 (100%)	78 (87.6%)	0.686
1	19 (11.2%)	3 (10.3%)	5 (10.6%)	0 (0%)	11 (12.4%)	
**Prior antibiotic course < 90 days**						
0	124 (72.9%)	22 (75.9%)	33 (70.2%)	4 (80.0%)	65 (73.0%)	0.872
1	46 (27.1%)	7 (24.1%)	14 (29.8%)	1 (20.0%)	24 (27.0%)	
**Myocardial infarction**						
0	143 (84.1%)	23 (79.3%)	43 (91.5%)	5 (100%)	72 (80.9%)	0.163
1	27 (15.9%)	6 (20.7%)	4 (8.5%)	0 (0%)	17 (19.1%)	
**Congestive heart failure**						
0	165 (97.1%)	28 (96.6%)	47 (100%)	5 (100%)	85 (95.5%)	0.3
1	5 (2.9%)	1 (3.4%)	0 (0%)	0 (0%)	4 (4.5%)	
**Vascular peripheral disease**						
0	161 (94.7%)	27 (93.1%)	45 (95.7%)	5 (100%)	84 (94.4%)	0.822
1	9 (5.3%)	2 (6.9%)	2 (4.3%)	0 (0%)	5 (5.6%)	
Stroke						
0	162 (95.3%)	29 (100%)	47 (100%)	5 (100%)	81 (91.0%)	0.0839
1	8 (4.7%)	0 (0%)	0 (0%)	0 (0%)	8 (9.0%)	
**Dementia**						
0	168 (98.8%)	29 (100%)	47 (100%)	4 (80.0%)	88 (98.9%)	0.0014
1	2 (1.2%)	0 (0%)	0 (0%)	1 (20.0%)	1 (1.1%)	
**Chronic respiratory failure**						
0	148 (87.1%)	26 (89.7%)	41 (87.2%)	5 (100%)	76 (85.4%)	0.639
1	22 (12.9%)	3 (10.3%)	6 (12.8%)	0 (0%)	13 (14.6%)	
**Connective tissue disease**						
0	168 (98.8%)	29 (100%)	46 (97.9%)	5 (100%)	88 (98.9%)	0.862
1	2 (1.2%)	0 (0%)	1 (2.1%)	0 (0%)	1 (1.1%)	
**Gastric ulcer**						
0	167 (98.2%)	28 (96.6%)	47 (100%)	5 (100%)	87 (97.8%)	0.553
1	3 (1.8%)	1 (3.4%)	0 (0%)	0 (0%)	2 (2.2%)	
**Mild liver disease**						
0	166 (97.6%)	28 (96.6%)	46 (97.9%)	5 (100%)	87 (97.8%)	0.944
1	4 (2.4%)	1 (3.4%)	1 (2.1%)	0 (0%)	2 (2.2%)	
**Moderate to severe liver disease**						
0	168 (98.8%)	29 (100%)	47 (100%)	5 (100%)	87 (97.8%)	0.553
1	2 (1.2%)	0 (0%)	0 (0%)	0 (0%)	2 (2.2%)	
**Kidney disease**						
0	160 (94.1%)	26 (89.7%)	45 (95.7%)	5 (100%)	84 (94.4%)	0.822
1	10 (5.9%)	3 (10.3%)	2 (4.3%)	0 (0%)	5 (5.6%)	
**Diabetes—without complications**						
0	118 (69.4%)	23 (79.3%)	28 (59.6%)	4 (80.0%)	63 (70.8%)	0.344
1	52 (30.6%)	6 (20.7%)	19 (40.4%)	1 (20.0%)	26 (29.2%)	
**Diabetes—with complications**						
0	164 (96.5%)	28 (96.6%)	46 (97.9%)	5 (100%)	85 (95.5%)	0.707
1	6 (3.5%)	1 (3.4%)	1 (2.1%)	0 (0%)	4 (4.5%)	
**Solid cancer without metastasis < 5 years**						
0	148 (87.1%)	24 (82.8%)	42 (89.4%)	5 (100%)	77 (86.5%)	0.623
1	22 (12.9%)	5 (17.2%)	5 (10.6%)	0 (0%)	12 (13.5%)	
**Solid cancer with metastasis**						
0	166 (97.6%)	28 (96.6%)	46 (97.9%)	5 (100%)	87 (97.8%)	0.944
1	4 (2.4%)	1 (3.4%)	1 (2.1%)	0 (0%)	2 (2.2%)	
**Leukemia**						
0	167 (98.2%)	29 (100%)	46 (97.9%)	5 (100%)	87 (97.8%)	0.944
1	3 (1.8%)	0 (0%)	1 (2.1%)	0 (0%)	2 (2.2%)	
**Lymphoma**						
0	167 (98.2%)	28 (96.6%)	46 (97.9%)	5 (100%)	88 (98.9%)	0.862
1	3 (1.8%)	1 (3.4%)	1 (2.1%)	0 (0%)	1 (1.1%)	
**Hypertension**						
0	86 (50.6%)	10 (34.5%)	24 (51.1%)	2 (40.0%)	50 (56.2%)	0.695
1	84 (49.4%)	19 (65.5%)	23 (48.9%)	3 (60.0%)	39 (43.8%)	
**Tobacco consumption**						
0	126 (74.1%)	22 (75.9%)	36 (76.6%)	4 (80.0%)	64 (71.9%)	0.797
1	44 (25.9%)	7 (24.1%)	11 (23.4%)	1 (20.0%)	25 (28.1%)	
**COVID-19 wave**						
wave1	30 (17.6%)	6 (20.7%)	8 (17.0%)	0 (0%)	16 (18.0%)	0.244
wave2	51 (30.0%)	8 (27.6%)	17 (36.2%)	2 (40.0%)	24 (27.0%)	
wave3	72 (42.4%)	13 (44.8%)	14 (29.8%)	2 (40.0%)	43 (48.3%)	
wave4	17 (10.0%)	2 (6.9%)	8 (17.0%)	1 (20.0%)	6 (6.7%)	
**Charlson Score**						
Mean (SD)	4.59 (2.28)	4.79 (2.50)	4.30 (1.99)	3.20 (0.837)	4.76 (2.38)	0.137
Median [IQR]	4.00 [3.00–6.00]	5.00 [3.00–6.00]	4.00 [3.00–5.00]	3.00 [3.00–4.00]	4.00 [3.00–6.00]	
**SAPS II at ICU admission**						
Mean (SD)	41.1 (15.7)	41.1 (15.6)	43.9 (14.2)	29.4 (5.03)	40.3 (16.6)	0.0274
Median [IQR]	38.0 [29.0–49.0]	39.0 [29.0–50.0]	41.0 [33.0–50.0]	29.0 [29.0–31.0]	36.0 [29.0–46.0]	
**SOFA at ICU admission**						
Mean (SD)	5.09 (2.95)	4.45 (2.57)	5.68 (3.07)	4.00 (1.22)	5.06 (3.04)	0.366
Median [IQR]	4.00 [3.00–8.00]	3.00 [3.00–6.00]	4.00 [3.00–8.00]	4.00 [4.00–5.00]	4.00 [2.00–8.00]	
**Prophylactic antibiotic therapy**						
0	102 (60.0%)	16 (55.2%)	25 (53.2%)	3 (60.0%)	58 (65.2%)	0.395
1	68 (40.0%)	13 (44.8%)	22 (46.8%)	2 (40.0%)	31 (34.8%)	
**Dexamethasone**						
0	32 (18.8%)	6 (20.7%)	9 (19.1%)	0 (0%)	17 (19.1%)	0.557
1	138 (81.2%)	23 (79.3%)	38 (80.9%)	5 (100%)	72 (80.9%)	
**Vasopressors**						
0	91 (53.5%)	20 (69.0%)	14 (29.8%)	2 (40.0%)	55 (61.8%)	0.0016
1	79 (46.5%)	9 (31.0%)	33 (70.2%)	3 (60.0%)	34 (38.2%)	
**Mechanical ventilation**						
0	21 (12.4%)	6 (20.7%)	3 (6.4%)	1 (20.0%)	11 (12.4%)	0.442
1	149 (87.6%)	23 (79.3%)	44 (93.6%)	4 (80.0%)	78 (87.6%)	
**Meduri corticosteroids protocol ****						
0	146 (85.9%)	27 (93.1%)	40 (85.1%)	5 (100%)	74 (83.1%)	0.592
1	24 (14.1%)	2 (6.9%)	7 (14.9%)	0 (0%)	15 (16.9%)	
**Hydroxychloroquine**						
0	147 (86.5%)	21 (72.4%)	43 (91.5%)	5 (100%)	78 (87.6%)	0.578
1	23 (13.5%)	8 (27.6%)	4 (8.5%)	0 (0%)	11 (12.4%)	
**Lopinavir/ritonavir**						
0	156 (91.8%)	26 (89.7%)	43 (91.5%)	5 (100%)	82 (92.1%)	0.796
1	14 (8.2%)	3 (10.3%)	4 (8.5%)	0 (0%)	7 (7.9%)	
**Anti-IL1 (Kineret)**						
0	165 (97.1%)	29 (100%)	44 (93.6%)	5 (100%)	87 (97.8%)	0.421
1	5 (2.9%)	0 (0%)	3 (6.4%)	0 (0%)	2 (2.2%)	
**Anti-JAK2 (Jakavi)**						
0	162 (95.3%)	28 (96.6%)	43 (91.5%)	5 (100%)	86 (96.6%)	0.369
1	8 (4.7%)	1 (3.4%)	4 (8.5%)	0 (0%)	3 (3.4%)	
**Tocilizumab**						
0	134 (78.8%)	24 (82.8%)	35 (74.5%)	4 (80.0%)	71 (79.8%)	0.441
1	4 (2.4%)	1 (3.4%)	2 (4.3%)	0 (0%)	1 (1.1%)	
Missing	32 (18.8%)	4 (13.8%)	10 (21.3%)	1 (20.0%)	17 (19.1%)	
**In-hospital immunosuppressors**						
0	132 (77.6%)	26 (89.7%)	33 (70.2%)	4 (80.0%)	69 (77.5%)	0.368
1	35 (20.6%)	3 (10.3%)	13 (27.7%)	0 (0%)	19 (21.3%)	
Missing	3 (1.8%)	0 (0%)	1 (2.1%)	1 (20.0%)	1 (1.1%)	
**Acquired resistance—Overall *****						
0	161 (94.7%)	29 (100%)	42 (89.4%)	4 (80.0%)	86 (96.6%)	0.115
1	9 (5.3%)	0 (0%)	5 (10.6%)	1 (20.0%)	3 (3.4%)	
**Rectal swab MDR**						
0	167 (98.2%)	29 (100%)	45 (95.7%)	5 (100%)	88 (98.9%)	0.458
1	3 (1.8%)	0 (0%)	2 (4.3%)	0 (0%)	1 (1.1%)	
**Acquired resistance in other samples**						
0	162 (95.3%)	29 (100%)	43 (91.5%)	4 (80.0%)	86 (96.6%)	0.173
1	8 (4.7%)	0 (0%)	4 (8.5%)	1 (20.0%)	3 (3.4%)	
**Total duration of antibiotic treatment (days)**						
Mean (SD)	8.86 (28.5)	18.6 (67.4)	10.2 (10.6)	12.0 (5.34)	4.82 (3.05)	0.001
Median [IQR]	6.00 [3.00–7.00]	7.00 [6.00–7.00]	7.00 [6.00–9.00]	10.0 [10.0–12.0]	4.00 [3.00–7.00]	
**Relapse of infection**						
0	155 (91.2%)	26 (89.7%)	41 (87.2%)	3 (60.0%)	85 (95.5%)	0.0095
1	14 (8.2%)	2 (6.9%)	6 (12.8%)	2 (40.0%)	4 (4.5%)	
Missing	1 (0.6%)	1 (3.4%)	0 (0%)	0 (0%)	0 (0%)	
**Recurrence of infection**						
0	143 (84.1%)	26 (89.7%)	34 (72.3%)	2 (40.0%)	81 (91.0%)	0.001
1	26 (15.3%)	2 (6.9%)	13 (27.7%)	3 (60.0%)	8 (9.0%)	
Missing	1 (0.6%)	1 (3.4%)	0 (0%)	0 (0%)	0 (0%)	
**Mechanical ventilation (days)**						
Mean (SD)	21.9 (16.5)	15.7 (9.77)	24.4 (15.8)	45.8 (24.7)	21.4 (17.1)	0.0547
Median [IQR]	17.0 [11.0–30.0]	13.0 [8.00–19.0]	20.0 [12.0–36.0]	47.5 [37.3–56.0]	16.0 [11.0–27.0]	
Missing	13 (7.6%)	2 (6.9%)	2 (4.3%)	1 (20.0%)	8 (9.0%)	
**ICU length of stay (d)**						
Mean (SD)	27.7 (19.7)	18.2 (10.1)	33.3 (21.9)	43.4 (30.3)	26.9 (18.9)	0.183
Median [IQR]	21.0 [15.0–35.5]	15.0 [12.0–19.0]	25.0 [18.5–46.5]	52.0 [15.0–68.0]	21.0 [15.0–31.0]	
**Hospital length of stay (d)**						
Mean (SD)	35.1 (21.3)	24.3 (12.8)	43.3 (24.7)	49.6 (29.3)	33.4 (19.3)	0.0562
Median [IQR]	29.5 [20.0–46.0]	20.0 [17.0–28.0]	37.0 [24.5–54.5]	60.0 [23.0–75.0]	30.0 [20.0–45.0]	
**Sepsis to ICU discharge (d)**						
Mean (SD)	28.1 (20.3)	18.6 (10.1)	34.1 (23.1)	43.4 (30.9)	27.1 (19.4)	0.159
Median [IQR]	21.0 [15.0–35.5]	17.0 [12.0–20.0]	25.0 [18.5–46.5]	52.0 [15.0–67.0]	21.0 [15.0–32.0]	
**Sepsis to hospital discharge (d)**						
Mean (SD)	34.7 (23.7)	22.8 (12.9)	43.5 (26.3)	49.4 (29.6)	33.1 (22.8)	0.0393
Median [IQR]	29.0 [19.0–43.8]	19.0 [14.0–26.0]	37.0 [25.0–55.5]	60.0 [23.0–75.0]	29.0 [18.0–43.0]	
**Withdrawal or withholding of care**						
0	156 (91.8%)	25 (86.2%)	45 (95.7%)	5 (100%)	81 (91.0%)	0.486
1	14 (8.2%)	4 (13.8%)	2 (4.3%)	0 (0%)	8 (9.0%)	
**ICU mortality**						
0	124 (72.9%)	17 (58.6%)	41 (87.2%)	4 (80.0%)	62 (69.7%)	0.0729
1	46 (27.1%)	12 (41.4%)	6 (12.8%)	1 (20.0%)	27 (30.3%)	
**In-hospital mortality**						
0	123 (72.4%)	17 (58.6%)	40 (85.1%)	4 (80.0%)	62 (69.7%)	0.136
1	47 (27.6%)	12 (41.4%)	7 (14.9%)	1 (20.0%)	27 (30.3%)	
**Day 28 Mortality**						
0	124 (72.9%)	17 (58.6%)	40 (85.1%)	4 (80.0%)	63 (70.8%)	0.174
1	46 (27.1%)	12 (41.4%)	7 (14.9%)	1 (20.0%)	26 (29.2%)	
**Day 90 Mortality**						
0	123 (72.4%)	17 (58.6%)	40 (85.1%)	4 (80.0%)	62 (69.7%)	0.136
1	47 (27.6%)	12 (41.4%)	7 (14.9%)	1 (20.0%)	27 (30.3%)	

*SD:* standard deviation, *IQR:* interquartile range, *BMI*: body mass index, *ICU*: intensive care unit, *MDR*: multidrug-resistant, *SOFA:* Sepsis-related Organ Failure Assessment, *SAPS II*: Simplified Acute Physiology Score, *AIDS:* acquired immunodeficiency syndrome, *(d):* days; Charlson score: age > 40 years, myocardial infarction, congestive heart failure, vascular peripheral disease, stroke, dementia, respiratory insufficiency, rheumatologic disease, gastric ulcer, liver disease, kidney disease, diabetes, central palsy, solid cancer with or without metastasis, leukemia, lymphoma, AIDS; * Immunosuppression: treatment with corticosteroid, chemotherapy, immunotherapy, or other immunosuppressive drugs, history of transplant, blood disease resulting in immunosuppression; ** Meduri protocol [17]; *** Overall acquired resistance was defined as the appearance of an MDR bacteria either in rectal swabs or other samples.

**Table 2 antibiotics-11-01517-t002:** Motivations for ADE in the COVID-19 group.

	n (%)
Replacement of an empirical antimicrobial agent with a narrower-spectrum antibiotic	29/47 (61.7)
Stopping one or more components of an empirical combination therapy	18/47 (38.2)
Cases of microbiologically confirmed infection, causative pathogen is covered by concomitant antimicrobial therapy	5/18
Cases of non-microbiologically confirmed infection	13/18

Results are shown as *n (%).*

## Data Availability

Restrictions apply to the availability of these data. Data was obtained from the Assistance Publique des Hôpitaux de Marseille and are available from the authors with the permission of the Assistance Publique des Hôpitaux de Marseille.

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
