# Peer review of "Antimicrobial Stewardship during COVID-19 Outbreak: A Retrospective Analysis of Antibiotic Prescriptions in the ICU across COVID-19 Waves"

_antibiotics, 2022, doi:10.3390/antibiotics11111517_

Round 1

Reviewer 1 Report

This study describes the antibiotic stewardship practices during COVID-19 outbreak in three ICU wards from a French University hospital, focusing on the appropriateness of empirical antibiotic treatments and antibiotic deescalation habits.

The manuscript is well written and results are clearly stated. It is interesting to have access to a comprehensive view of antibiotic strategies from the authors’ institution, despite missing datas inherent to the retrospective design of the study.

However, I find it very unfortunate to only compare these datas to datas from the literature: as stated by the authors, antibiotic stewardship habits are very heterogeneous depending on teams/institutions (e.g. regarding ADE rates, they cite  rates between 16 and 81% in the discussion part). Therefore the reader cannot aprehend the real impact of COVID-19 outbreak on the local antimicrobial stewardship habits (ADE rates, empirical antibiotic treatment protocols, resistance rates…), which seemed to be the main point of the study. Maybe without going into so much details the authors could add some retrospective data concerning antibiotic stewardship in non-COVID-19 patients suffering from acute respiratory failure in 2018/2019: it would provide the reader with local ADE rates, mortality rates, or MDR bacteria numbers, and would greatly improve the discussion part.

Other comments may be found below:

-          Line 239. The authors distinguish between antibiotic “prophylactic treatment” and “empirical treatment”. What was the “prophylaxis” for? A prophylaxis is supposed to prevent an infection (e.g. azithromycin preventing MAC infections in HIV patients), what infection did they aim to prevent? Was not the 3GC+azithromycin treatment supposed to target a potential bacterial superinfection during the SARS Cov2 infection (in addition to a potential antiinflammatory/antiviral effect of azithromycin which has sometimes be described in the literature?) Moreover, this “prophylactic” treatment is not mentioned elsewhere in the manuscript (e.g. in results, Table 1, Figure 1,eTables…); to whom  was it given? Please explain and add this information in Figure 1/Tables  if necessary

-          Table 1. Please add (N=141) to the “empirical antibiotic treatment” headline

-          Table 1. “acquired resistance”. Does it mean patients were negative for MDR bacteria in initial samples, and that samples from the same sampling sites were then found positive for MDR bacteria? Please add details (as a footnote or in the method part). Similarly, at line 106, does the 0.05% rate refer to conversion of recatal swabs from negative to positive during hospitalization?

-          Table 1. “rectal swab MDR”. As rectal swabing was performed every week (line 260), does it mean patients were found positive at least at one sampling date? Or does it mean patients were negative at arrival and then found positive after ATB treatment? Please explain (as a footnote or in the method part)

-          Figure 1. There are 29 patients “without any empirical treatment”. Does it mean they belong to the “targeted ATB treatment group” mentioned in the method part (I believe so since in Table 1 there is also a treatment duration for this group)? Please keep the same group names throughout the manuscript for a better understanding.

-          eTable 3: As most bacterial findings come from respiratory samples, is there more information relative to the method  of sampling (BAL, lung aspirate, …)?

-          eTAble 3: It is strange that all Enterobacter spp were ampicillin-S, this is a rare feature in this genera which commonly produces a chromosomal AmpC betalactamase

-          Only antibacterial agents are mentioned in this “antimicrobial stewardship” during COVID-19 outbreak. Some authors in various countries mentioned superinfections due to Aspergillus spp and Pneumocystis jirovecii in COVID-19 patients, did the authors looked for these pathogens? Were there empirical/targeted therapies for these pathogens (especially when it comes to Pneumocystis jirovecii which is treated using high-dose antibiotics which may impact the emergence of MDR bacteria or the outcome of infections?)

-          Line 86. What was the most frequent ATB association among these 16% cases (e.g. beta-lactam+aminoglycoside, or beta-lactam + anti-Gram positive drug, …)?

-          Line 103 and Figure 1. Among “documented infections”, the empirical treatment did not cover the identified bacteria in 10/52 patients. However, in Figure 1, escalation was performed only in 4 patients. Therefore 6 patients kept receiving on purpose antibiotics which did not target the identified pathogen? Please explain.

-          Line 93. An eTable detailing the 30 “antibiotic de-escalations “ would be welcomed, as there is no mention of what the authors call “wide spectrum” and “narrow spectrum” ATB in the present manuscript  (and one knows that the various definitions of “wide spectrum” and “narrow spectrum” antibiotics is subject to debates !)

-          Line 102. It would be interesting to add in Figure 1 the notion of “documented infection”, to know the respective numbers of “documented infections” amongst each group (deescalation, escalation, continuation)

-          There are some rare typo mistakes throughout the manuscript, please re-read carefully (e.g. eTable2, Bacteriemia=>Bacteremia; eTable 3: H. influezae => H. influenzae)

Author Response

Dear Reviewer 1,

Thank you for your constructive comments which have highlighted several important issues that required amendment and clarification. We hope that the revised version is more clear and meets expectations.

However, I find it very unfortunate to only compare these data to data from the literature: as stated by the authors, antibiotic stewardship habits are very heterogeneous depending on teams/institutions (e.g. regarding ADE rates, they cite  rates between 16 and 81% in the discussion part). Therefore the reader cannot apprehend the real impact of COVID-19 outbreak on the local antimicrobial stewardship habits (ADE rates, empirical antibiotic treatment protocols, resistance rates…), which seemed to be the main point of the study. Maybe without going into so much details the authors could add some retrospective data concerning antibiotic stewardship in non-COVID-19 patients suffering from acute respiratory failure in 2018/2019: it would provide the reader with local ADE rates, mortality rates, or MDR bacteria numbers, and would greatly improve the discussion part.

The comparison with a historical control group has been done as wisely suggested. It has yielded really interesting results, we therefore are greatly grateful to you to have helped us improve our data and results. We decided to look across the years 2018 and 2019, with a time constraint of three months. This sample population is in line with analyses previously published by our team on ADE (reference n°22 in the reference list).

Here are the main results for this analysis:

Control group

Sixty-seven patients were included in the control group (e-Table 6; e-Figure 1). The main differences between the two groups were the body mass index (BMI) (30±5.8 in COVID-19 vs 24.5±4.9 in control group, p = 0.001), previous hospitalization (11.2% in COVID-19 vs 49.3% in control group, p = 0.01) and previous antibiotic treatment in the 90 days (27.1% in COVID-19 vs 46.3% in control group, p = 0.001) (e-Table 7). The control group had higher SAPSII than the COVID-19 group (49±16 vs. 41±16, p < 0.001) (e-Table 7). Among those receiving an empirical antibiotic treatment (n = 58), empirical antibiotic treatment was de-escalated, escalated, and continued in 34 (58.6%), 3 (5%), and 21 (36.2%) patients, respectively (e-Figure 1).

The ADE rate was 52.2% in the control group and 27.6% in the COVID-19 group (p < 0.001). The rates of acquired resistance was higher in the control group than in the COVID-19 group (16.4% vs. 5.3%, p = 0.009). No differences were reported for the ICU, day-28 and day-90 mortality rates between the two groups (e-Table 7).”

Of note, there were changes in some figures regarding the COVID-19 cohort as the data was thoroughly revised by the authors while creating the historical cohort.

Other comments may be found below:

-          Line 239. The authors distinguish between antibiotic “prophylactic treatment” and “empirical treatment”. What was the “prophylaxis” for? A prophylaxis is supposed to prevent an infection (e.g. azithromycin preventing MAC infections in HIV patients), what infection did they aim to prevent? Was not the 3GC+azithromycin treatment supposed to target a potential bacterial superinfection during the SARS Cov2 infection (in addition to a potential antiinflammatory/antiviral effect of azithromycin which has sometimes be described in the literature?) Moreover, this “prophylactic” treatment is not mentioned elsewhere in the manuscript (e.g. in results, Table 1, Figure 1,eTables…); to whom  was it given? Please explain and add this information in Figure 1/Tables  if necessary

We thank you for this comment. Antibiotic prophylaxis is detailed and explained in the Method section as follows: “Antibiotic prophylaxis consisted of the administration of prophylactic antibiotics at ICU admission before the suspicion of bacterial infection (this prophylaxis mainly combined a third-generation cephalosporin with azithromycin and was mostly prescribed during the first wave)” . To add further details and clarify the meaning of this definition, the following sentence has been added: “It was intended to prevent potential bacterial superinfection during Sars-CoV-2 infection and was systematically administered on admission to ICU to all COVID-19 patients in the first wave. This practice was gradually discontinued over the waves.”

Furthermore, results regarding this prophylaxis has been added in the Table 1: there was no difference between groups regarding its administration.

-          Table 1. Please add (N=141) to the “empirical antibiotic treatment” headline

The modification has been done accordingly.

-          Table 1. “acquired resistance”. Does it mean patients were negative for MDR bacteria in initial samples, and that samples from the same sampling sites were then found positive for MDR bacteria? Please add details (as a footnote or in the method part). Similarly, at line 106, does the 0.05% rate refer to conversion of rectal swabs from negative to positive during hospitalization?

Thank your comment. The following sentence has been added to both the method section and the footnotes to clarify this issue: “Overall acquired resistance was defined as the appearance of a MDR bacteria either in rectal swabs or othersamples.” As for the 0.05% rate, the following sentence has been added as well: “found either in rectal swabs or other samples, Table 1”.

-          Table 1. “rectal swab MDR”. As rectal swabbing was performed every week (line 260), does it mean patients were found positive at least at one sampling date? Or does it mean patients were negative at arrival and then found positive after ATB treatment? Please explain (as a footnote or in the method part)

Thank you for this comment. The following sentence has been added to the method section “these tests were repeated regardless of whether the first or the following rectal swabs were positive”. Was also added the following information: “Patients who were already positive for MDR on admission were not considered to have acquired antimicrobial resistance, unless a new type of resistance or new MDR bacteria emerged during their stay in the ICU.” We hope it will help clarify the meaning.

-          Figure 1. There are 29 patients “without any empirical treatment”. Does it mean they belong to the “targeted ATB treatment group” mentioned in the method part (I believe so since in Table 1 there is also a treatment duration for this group)? Please keep the same group names throughout the manuscript for a better understanding.

We thank you for this question. The subsequent sentences have been added to the method section to further explain and define group names: “Patients who received targeted antibiotic treatment as first-line therapy were classified in the “no empirical treatment” group. Patients who received both empirical and secondarily targeted antibiotic treatment were classified in the “empirical treatment”.

-          eTable 3: As most bacterial findings come from respiratory samples, is there more information relative to the method of sampling (BAL, lung aspirate, …)?

Thank you for this interesting question. The method of sampling has now been added as e-Table 5.

-          eTAble 3: It is strange that all Enterobacter spp were ampicillin-S, this is a rare feature in this genera which commonly produces a chromosomal AmpC betalactamase

We thank you for spotting this mistake. The table has now been corrected and Enterobacter spp are cefotaxime-S.

-          Only antibacterial agents are mentioned in this “antimicrobial stewardship” during COVID-19 outbreak. Some authors in various countries mentioned superinfections due to Aspergillus spp and Pneumocystis jirovecii in COVID-19 patients, did the authors looked for these pathogens? Were there empirical/targeted therapies for these pathogens (especially when it comes to Pneumocystis jirovecii which is treated using high-dose antibiotics which may impact the emergence of MDR bacteria or the outcome of infections?)

We thank you for this question. Unfortunately we did not look to this data as the focus was on antibiotic de-escalation. We believe this is an important matter that should be studied in further studies. The following sentence has been added to the limitations section of our discussion “Other antimicrobial agents, such as antifungals and antivirals were not included in our analysis.”

Of note, in our ICU the rates of superinfection due to Aspergillus spp remained low.

-          Line 86. What was the most frequent ATB association among these 16% cases (e.g. beta-lactam+aminoglycoside, or beta-lactam + anti-Gram positive drug, …)?

The most frequent association was beta-lactam plus aminoglycosides and this has been added to the manuscript.

-          Line 103 and Figure 1. Among “documented infections”, the empirical treatment did not cover the identified bacteria in 10/52 patients. However, in Figure 1, escalation was performed only in 4 patients. Therefore 6 patients kept receiving on purpose antibiotics which did not target the identified pathogen? Please explain.

Thank you for this comment and for giving us the opportunity to amend an important and potentially misleading oversight. We examined thoroughly the data and corrected a mistake: 5/55 patients had an empirical treatment that did not cover the identified bacteria. All of them were in the escalation group. The following sentence has been added to clarify this issue: “Among the 55 patients with a documented infection, the empirical antibiotic treatment was appropriate, i.e. covering the identified bacteria, in 50 (90.9%) patients. The five patients who received inappropriate antibiotic treatment were in the escalation group.”

-          Line 93. An eTable detailing the 30 “antibiotic de-escalations “ would be welcomed, as there is no mention of what the authors call “wide spectrum” and “narrow spectrum” ATB in the present manuscript  (and one knows that the various definitions of “wide spectrum” and “narrow spectrum” antibiotics is subject to debates !)

This e-Table has now been added to the supplemental material (e-Table 6).

-          Line 102. It would be interesting to add in Figure 1 the notion of “documented infection”, to know the respective numbers of “documented infections” amongst each group (deescalation, escalation, continuation)

Thank you for this comment. Figure 1 has been modified accordingly.

-          There are some rare typo mistakes throughout the manuscript, please re-read carefully (e.g. eTable2, Bacteriemia=>Bacteremia; eTable 3: H. influezae => H. influenzae)

Thank you for this comment. The manuscript has been re-read scrupulously

We thank you for your attention to this paper, and we have scrupulously followed your advice and corrections. We hope that these improvements will bring this manuscript to the level of your expectations.

Sincerely yours,

Inès Lakbar

Reviewer 2 Report

This manuscript describes antimicrobial stewardship in the ICU patients with COVID-19. It looks interesting and beneficial for us. However there remains some points to be reconfirmed.

Major points

1)    If a patient had multiple bacterial infections during ICU stay, how did you count these episodes? Did you count all the episodes or just the first one?

2)    When did you perform ADE, at the time when causative organism was isolated? Please describe it.

3)    In page 3, you described the antibiotic treatment duration was 4.8 days in the continuation group, much shorter than the other groups. Does this group include the patients with discontinuation of all antibiotics? If so, I think some of these cases might be classified as ADE group.

4)    In page 17, why did you exclude dexamethasone administration from immunosuppressive treatments? You only used dexamethasone only a short time?

Minor points

1)    In page 7, what are inflammatory diseases? 

Author Response

This manuscript describes antimicrobial stewardship in the ICU patients with COVID-19. It looks interesting and beneficial for us. However there remains some points to be reconfirmed.

Dear Reviewer 2,

It  is our pleasure to respond to each of your comments. We thank you for this relevant review that allowed us to improve our manuscript.

Major points

  • If a patient had multiple bacterial infections during ICU stay, how did you count these episodes? Did you count all the episodes or just the first one?

Thank you for raising this matter. The following sentence has been added to clarify this issue : “In case of multiple bacterial infections during ICU stay, only the first one was considered.”

  • When did you perform ADE, at the time when causative organism was isolated? Please describe it. 

We thank you for this interesting question. The following sentence has been added to the manuscript “Antibiotic treatments were classified according to the first change in treatment (or lack thereof) between day 0 and the day the microbiological results were returned.” We hope this answers your relevant question.

  • In page 3, you described the antibiotic treatment duration was 4.8 days in the continuation group, much shorter than the other groups. Does this group include the patients with discontinuation of all antibiotics? If so, I think some of these cases might be classified as ADE group. 

Thank you for this question. As defined in the guidelines, discontinuation of all antibiotics should not be considered as ADE, we therefore classify these patients in the continuation group. We however agree with you that this is a bias and it could be misleading, thus we did an analysis excluding patients with discontinuation of all antibiotics and it yielded interesting results (p3, line 126): “Excluding patients who discontinued all antibiotics, the mean duration of antibiotic treatment in the continuation group increased to 7.3 (±2.8) days (n = 41)”

The following sentence has also been added to the discussion section: “Third, the patients whose antibiotics were stopped at the time of reporting microbiological results were classified in the continuation group, as the guidelines stated that early discontinuation of all antibiotics cannot be considered as ADE [29].  This could have led to bias in our analysis.”

  • In page 17, why did you exclude dexamethasone administration from immunosuppressive treatments? You only used dexamethasone only a short time? 

We wanted to separate immunosuppressive treatment used outside the scope of COVID-19 and those used specifically in this indication, such as the anti-IL1, anti-JAK2, anti-IL6 and dexamethasone, to allow the reader to have a detailed view of the treatments received in each group. 

Minor points 

1)    In page 7, what are inflammatory diseases? 

This is a typo mistake, inflammatory diseases has now been corrected and replaced by connective tissue disease as usually reported in the Charlson score.

We thank you for your time and attention, and we are grateful that you give us the opportunity to correct and improve our manuscript. We hope that these improvements will bring this protocol to the level of your expectations.

Sincerely yours,

Inès Lakbar

Reviewer 3 Report

Many thanks for this opportunity to read this interesting paper. The topic is very relevant and crucial as global heath problem and central in infectious diseases 

Below my suggestions

1. Introduction: Antimicrobial resistance is an urgent threat to public health and global development; in this scenario, the SARS-CoV2 pandemic has caused a major disruption of healthcare systems and practices. A narrative review was conducted on articles focusing on the impact of COVID-19 on multidrug-resistant gram-negative, gram-positive bacteria, and fungi. (see and cite this recent review Impact of SARS-CoV-2 Epidemic on Antimicrobial Resistance: A Literature Review. Viruses. 2021 Oct 20;13(11):2110. doi: 10.3390/v13112110.)

2.Methods and result: well presented

3. Discussion: In my opinion is important stress on this concept 

3a. the role of fast diagnostic microbiology to reduce empiric therapy

3b. the role of immunodepression in candidemia 

3b. in AMR contrast is crucial the education on this issue especially during medical school. Consider this suggestions.

: (i) the introduction of an AMR course within the medical degree programme and during residency programmes; (ii) the setting up of an AMS programme in health districts and hospitals; and (iii) the institution of a network on AMR, with the AMR sentinel doctors directly involved in monitoring and evaluating trends in AMR in their health districts and hospitals. (see Italian young doctors' knowledge, attitudes and practices on antibiotic use and resistance: A national cross-sectional survey. J Glob Antimicrob Resist. 2020 Dec;23:167-173. doi: 10.1016/j.jgar.2020.08.022.)

Conclusion: give some proposal that came from your interesting data 

Author Response

Dear Reviewer 3,

Many thanks for this opportunity to read this interesting paper. The topic is very relevant and crucial as global health problem and central in infectious diseases 

It  is our pleasure to respond to each of your comments. We thank you for this relevant review that allowed us to improve our manuscript.

Below my suggestions

  1. Introduction: Antimicrobial resistance is an urgent threat to public health and global development; in this scenario, the SARS-CoV2 pandemic has caused a major disruption of healthcare systems and practices. A narrative review was conducted on articles focusing on the impact of COVID-19 on multidrug-resistant gram-negative, gram-positive bacteria, and fungi. (see and cite this recent review Impact of SARS-CoV-2 Epidemic on Antimicrobial Resistance: A Literature Review. Viruses. 2021 Oct 20;13(11):2110. doi: 10.3390/v13112110.)

We thank you for this suggestion that allowed us to improve our introduction section. The sentence “The COVID-19 outbreak has caused a major disruption to healthcare systems and practices” has been added to the introduction section, and the paper by Segala et al has been cited as well.

2.Methods and result: well presented

We thank you for this comment.

  1. Discussion: In my opinion is important stress on this concept 

3a. the role of fast diagnostic microbiology to reduce empiric therapy

This point has been stressed as follows “The development of rapid diagnostic tests, such as multiplex PCR, may affect the occurrence of resistance to a greater extent than ADE by reducing the use of antibiotics”

3b. the role of immunodepression in candidemia 

The following sentence has been added to highlight the role of immunosuppression in the MDR emergence: “Finally, other factors may be involved in the emergence of MDR bacteria, notably immunosuppression”.

We focused only on bacterial infection and did not report the rates of candidemia. However, when looking at the records, only a few of the included patients developed candidemia.

3b. in AMR contrast is crucial the education on this issue especially during medical school. Consider this suggestions.

: (i) the introduction of an AMR course within the medical degree program and during residency programs; (ii) the setting up of an AMS program in health districts and hospitals; and (iii) the institution of a network on AMR, with the AMR sentinel doctors directly involved in monitoring and evaluating trends in AMR in their health districts and hospitals. (see Italian young doctors' knowledge, attitudes and practices on antibiotic use and resistance: A national cross-sectional survey. J Glob Antimicrob Resist. 2020 Dec;23:167-173. doi: 10.1016/j.jgar.2020.08.022.)

We thank you for this suggestion. We have corrected the manuscript according to your comment and added the following sentence : ”Hence, it is urgent to implement antimicrobial stewardship programs in all hospital training programs and during medical education [27].” The paper on Italian yound doctor’s knowledge was cited.

Conclusion: give some proposal that came from your interesting data 

The conclusion has been modified by the addition of this sentence: “The ADE rates were lower in the COVID-19 group than in the control group, suggesting that the stress associated with COVID-19 affected our practices”

We thank you for your time and attention, and we are grateful that you give us the opportunity to correct and improve our manuscript. We hope that these improvements will bring this manuscript to the level of your expectations.

Sincerely yours,

Inès Lakbar

Round 2

Reviewer 1 Report

The manuscript has been greatly improved when compared to its first version. Adding the pre-pandemic control group really brings new lights to antimicrobial stewardship behaviors observed during the pandemic in this institution.

I think there remains a minor error in the abstract:

-Line 32-34. The authors state that the ADE rate was similar before and after the pandemic, while they cite a significative difference at line 31 (52% versus 28%, p<0.001)? It seems to be an error in the abstract since they state the contrary at lines 181 and  363.

And maybe one point could be further discussed:

-Line 202-215. One of the commonly highlighted “benefits”  of antimicrobial stewardship programs is their interest in  preventing the emergence of acquired drug resistance. However, in the present study, the authors show  a higher ADE rate in the control group but also a higher rate of acquired resistance. What could explain these (at first view) conterintuitive data? The authors evoke immunosuppression (line 212), were immunosuppression rates higher in the control group? Could the higher load of antibiotics received before the present episode (“previous antibiotic treatment”, line 141) be part of the explanation (it would mean that global duration of antibiotic treatment rather than drug spectrum could be partly involved in the emergence of resistance)?

Author Response

Manuscript n° antibiotics-1921952 – Round 2

Marseille, October the 25th, 2022

Object: Response to Reviewers

Dear Editor, dear Reviewers,

We thank you for your comments and your questions. Please find our response below.

*Reviewer 1

Dear Reviewer 1,

We thank you for your time and attention, and we are grateful that you give us the opportunity to correct and improve our manuscript.

The manuscript has been greatly improved when compared to its first version. Adding the pre-pandemic control group really brings new lights to antimicrobial stewardship behaviors observed during the pandemic in this institution.

I think there remains a minor error in the abstract:

-Line 32-34. The authors state that the ADE rate was similar before and after the pandemic, while they cite a significative difference at line 31 (52% versus 28%, p<0.001)? It seems to be an error in the abstract since they state the contrary at lines 181 and  363.

Thank you for spotting this mistake, it has now been corrected: “The ADE rates were lower in the COVID-19 group than in the control group, suggesting that the stress associated with COVID-19 affected our practices”

And maybe one point could be further discussed:

-Line 202-215. One of the commonly highlighted “benefits”  of antimicrobial stewardship programs is their interest in  preventing the emergence of acquired drug resistance. However, in the present study, the authors show  a higher ADE rate in the control group but also a higher rate of acquired resistance. What could explain these (at first view) conterintuitive data? The authors evoke immunosuppression (line 212), were immunosuppression rates higher in the control group? Could the higher load of antibiotics received before the present episode (“previous antibiotic treatment”, line 141) be part of the explanation (it would mean that global duration of antibiotic treatment rather than drug spectrum could be partly involved in the emergence of resistance)?

Thank you for this very interesting comment. The following sentence has now been added to the manuscript: “Our data showed a higher ADE rate and unexpectedly a higher rate of acquired resistance in the control group than in the COVID-19 group. This could be explained by differences in baseline features between the two groups, particularly in terms of MDR risk factors (previous hospitalization, previous antibiotic treatment) and severity scores upon ICU admission, reflecting higher rates of frailty and exposure to MDR in the control group”

We thank you once again for your reviewing, we hope we have answered all your queries and comments.

Sincerely yours,

Inès Lakbar